# Peculiar Morphologies Obtained for 80/20 PLA/PA11 Blend with Small Amounts of Fumed Silica

**DOI:** 10.3390/nano11071721

**Published:** 2021-06-29

**Authors:** Damien Rasselet, Monica Francesca Pucci, Anne-Sophie Caro-Bretelle, José-Marie Lopez-Cuesta, Aurélie Taguet

**Affiliations:** 1Polymers Composites and Hybrids (PCH), IMT Mines Ales, 30319 Ales, France; damien.rasselet@mines-ales.fr (D.R.); Jose-marie.lopez-cuesta@mines-ales.fr (J.-M.L.-C.); 2LMGC, IMT Mines Ales, University Montpellier, CNRS, 30319 Ales, France; monica.pucci@mines-ales.fr (M.F.P.); anne-sophie.caro-bretelle@mines-ales.fr (A.-S.C.-B.)

**Keywords:** silica nanoparticles, PLA, PA11, polymer blend, microstructure, rheology

## Abstract

This work highlights the possibility of obtaining peculiar morphologies by adding fumed silica into 80/20 polylactic acid/polyamide11 (PLA/PA11) blends. Two kinds of fumed silica (A200 and trimethoxyoctylsilane modified R805 fumed silica) were dispersed (by twin-screw extrusion, TSE) at a weight amount of 5% in neat PLA, neat PA11 and a 80/20 PLA/PA11 blend. Thermal Gravimetric Analysis (TGA) was used to verify this 5 wt % amount. Oscillatory shear rheology tests were conducted on all the formulations: (1) on neat polymer nanocomposites (PLASi5, PLASiR5, PA11Si5, PA11SiR5); and (2) on polymer blend nanocomposites (PLA80Si5 and PLA80SiR5). Scanning Electron Microscope (SEM), Scanning Transmission Electron Microscope (STEM), Atomic Force Microscopy (AFM) characterizations and laser granulometry were conducted. Microscopic analysis performed on polymer blend nanocomposites evidenced a localization of A200 silica in the PA11 dispersed phase and R805 silica at the PLA/PA11 interface. Frequency sweep tests on neat polymer nanocomposites revealed a pronounced gel-like behavior for PLASi5 and PA11SiR5, evidencing a high dispersion of A200 in PLA and R805 in PA11. A yield behavior was also evidenced for both PLA80Si5 and PLA80SiR5 blends. For the blend nanocomposites, PA11 dispersed phases were elongated in the presence of A200 silica and a quasi-co-continuous morphology was observed for PLA80Si5, whereas PLA80SiR5 exhibits bridges of silica nanoparticles between the PA11 dispersed phases.

## 1. Introduction

Since the discovery of the plastic continent, use of polymers is becoming controversial due to their extraction and their uncontrolled end of life. Especially the pollution by non-biodegradable plastics, due to a lack of recycling possibilities, degradation into microplastics in the landscape are problematic. However, plastics have numerous advantages: they can save lives when they are used as COVID-19 protection clothes or any medical devices; they can lighten structures and then reduce the kerosene or oil consumption, decreasing the CO_2_ emission in the transportation domain. Polymers are also used in energy domain (fuel cells, battery, etc.) for devices with less CO_2_-emitting conversion principles.

In order to meet the environmental and social concerns, a way is to use bio-based and/or bio-degradable polymers. Bio-based allows to dispense from petroleum, whereas biodegradable prevents from the pollution of soils and seas. However, this last case is only recommended for special applications such as packaging, whereas for many other applications, biodegradability is not required.

PLA and PA11 are two bio-based polymers and their blend leads to a polymeric material with interesting properties, depending on the ratio of each polymer in the final blend. Indeed, it is now well-established that by varying the proportion of each polymer in a blend, it is possible to tune the final properties, especially fire [1] and mechanical [2] properties. The influence of the proportion of each polymer in PLA/PA11 blends was studied regarding the morphology [3], rheological properties [4], and mechanical properties [5].

Nanoparticles are particles with at least one dimension lower than 100 nm. Due to their high specific surface area, nanoparticles are known to improve mechanical, thermal, fire, barrier, or electrical properties of the polymeric medium in which they are dispersed. However, dispersion of nanoparticles in polymers must be controlled in order to take into account their high specific surface area. By dispersing nanoparticles into polymer blends it is possible to change the morphology and then the final properties. Parameters that play a key role in the morphology of polymer blend nanocomposites are nanoparticle parameters (size, shape, aspect ratio, surface chemistry), polymer parameters (viscosity and viscosity ratio, interfacial tension, melting temperature), and processing parameters (mainly sequence of mixing) [6]. Hence, by adding nanoparticles into a polymer blend it is possible to attain unexpected morphologies and enhanced final properties.

Regarding the influence of nanoparticles (NPs) on the morphology of polymer blend, NPs have a general tendency to extend the dual continuity zone [7,8,9,10,11]. The co-continuous morphology is a non-equilibrium morphology which is formed during the processing when the breakup time of the elongated domains becomes longer than coalescence phenomenon; that is to say when the relaxation processes are slowed down. Hence, when dispersed in the minor phase, NPs can slow down the relaxation of this phase and extend the dual continuity zone. This was explained by Steimann et al. [12] while adding glass spheres of 250 nm diameter into a poly(methyl methacrylate)/ polystyrene (PMMA/PS) blend. Liu et al. [13] explored the shape relaxation processes of PS droplets filled with nanosilica into a polypropylene (PP) matrix. Nano-silica were found to retard the relaxation process of the elongated PS droplets due to the increased friction that decreases the mobility of molecular chains. In another article, Wu et al. [14] observed a transition from sea-island to co-continuous morphology when adding small amounts of carbon black (CB) into a 80/20 Acrylonitrile-Butadiene-Styrene/polyamide6 (ABS/PA6) blend. They explained this change in morphology by the self-networking of CB into PA6 dispersed phase.

The increased elasticity and viscosity of the phase that hosted the nanoparticles influence the final morphology by slowing down the relaxation of elongated polymer phases during processing. Another parameter that influences the elongation and relaxation of the filled droplets is the interfacial tension [8]. Indeed, interfacial tension between two polymers is affected when nanoparticles are added in one of the two phases.

A previous study dealt with the dispersion of organoclay, sepiolite and carbon nanotubes (CNTs) into 90/10 and 70/30 PLA/PA11 blends [15]. The authors evidenced that the three fillers bearing different high aspect ratio preferentially located in the PA11 dispersed phase. Moreover, the three fillers were able to convert the droplet-matrix morphology into a co-continuous one. The sepiolite needles were more effective than the CNTs in inducing co-continuity, suggesting that the deformability of the nanoparticles was another important parameter to take into account. Fumed silica were used by Chen et al. [16] to refine the morphology of 80/20 PS/PP blends and to compatibilize this blend. However, to the best of our knowledge, the effect of different fumed silica (bearing different surface energy) on the morphology of PLA/PA11 blends was not studied.

In the present work, we propose to study the effect of spherical silica nanoparticles bearing a fractal structure (fumed silica) [17] on the final morphology of a 80/20 PLA/PA11 blend. We tested two different fumed silica: one hydrophilic and one hydrophobic. The microstructure of the blend nanocomposites is deeply investigated and a link is established between these microstructures and the rheological behavior.

## 2. Materials and Methods

### 2.1. Materials

The polylactide (PLA grade 3251D) used in this study was purchased from NatureWORKS (Minnetonka, MN, USA). It is a semi-crystalline grade. The polyamide 11 (PA11 grade LMFO) was produced by Arkema (Colombes, France) under the trade name Rilsan^®^. PLA and PA11 polymers exhibit a zero shear viscosity of 95 and 330 Pa.s at 210 °C, respectively. Aerosil A200 and R805 fumed silica nanoparticles were purchased from Evonik. Aerosil A200 is known to be hydrophilic without any surface modification whereas Aerosil R805 was modified with trimethoxyoctylsilane, in order to increase its hydrophobicity (48% of the surface is covered by octyl groups). All characteristics of silica are summarized in Table 1.

### 2.2. Nanocomposite Processing

Different kinds of formulations were prepared: (1) neat polymer nanocomposites: A200 and R805 at an amount of 5 wt % were incorporated in each pure PLA and PA11 polymers; (2) polymer blend nanocomposites: blends based on 80 wt % of PLA, 20 wt % of PA11 and 5 wt % of each silica were prepared. Either neat or blend nanocomposites were prepared in a co-rotating twin-screw extruder (Clextral BC21, France), with a screw length L of 1200 mm, a diameter D f 25 mm and a L/D ratio of 48. A vacuum pump was used to avoid oxidation and hydrolytic degradation during extrusion. For all formulations, polymer pellets were introduced in the feed zone (zone 1) whereas silica nanoparticles were introduced in the fifth zone (the screw contains 13 zones). The extrusion temperature profile at the feed zone was 60 °C for the neat polymers with silica and 80 °C for the filled blends. The extrusion temperature for all the other zones and the die was 180 °C (for PLA with silica) or 210 °C for the other formulations. A feeding rate of 6 kg/h and a 250 rpm screw speed were applied for the neat polymer nanocomposites. Whereas, a feeding rate of 4 kg/h and a 200 rpm screw speed were applied for the polymer blend nanocomposites. Prior to extrusion, PLA and PA11 pellets were dried overnight at 80 °C under vacuum; A200 and R805 silica powders were dried overnight at 110 °C in an oven. All extrusion conditions and composition of the formulations are summarized in Table 2 and Table 3.

All formulations mentioned in Table 3 were dried one night at 80 °C under vacuum prior to injection molded. Injection molding was performed using a Zamak Mercator mini-press to obtain disks with a diameter of 25 mm and a thickness of 1.5 mm.

### 2.3. Thermo-Gravimetric Analysis

Thermogravimetric analysis (TGA) were conducted on PLASi5, PLASiR5, PA11Si5, and PA11SiR5, mainly to verify the amount of silica in each neat polymer nanocomposite. Thermal characterization was carried out with a PerkinElmer Pyris-1 instrument on 10 mg of samples, under nitrogen. Samples were heated at 60 °C/min from 30 to 800 °C. Three tests were carried out for each nanocomposite.

### 2.4. Rheological Measurements

The rheology was performed using a MCR 702 rotational rheometer (Anton Paar, Austria) equipped with a parallel plate geometry (diameter of 25 mm). Tests were conducted at 210 °C under nitrogen with a gap of 1 mm into disks samples. All samples were dried under vacuum at 80 °C for one night prior to each test. Three tests were carried out for each protocol to ensure repeatability of the measurements. Error bars were then measured and added to any results.

To avoid any degradation, especially polyamide polycondensation [18], all the tests were conducted during a lower duration than 60 min.

The linear zone was determined with strain sweep tests conducted at 6.28 rad/s. The strain range goes from 0.01% to 50%. This protocol allows to determine the linear zone (i.e., the range of strains that keeps each sample in a linear response regarding the measured complex viscosity $\eta^{*}$). Regarding the curves obtained by this protocol, the linear range is up to 7% for all the formulations, except for PLA80SiR5. Hence, the next frequency sweep test for PLA80SiR5 must be conducted at a lower strain than 2%.

Finally, frequency sweep tests were conducted at a strain of 2% for all the formulations except for PLA80SiR5 where a 0.8% of strain was applied. The range of frequency goes from ω = 100 to 0.01 rad/s.

The choice of each parameter regarding those protocols are discussed in the Results and Discussion section.

### 2.5. Microstructure Characterizations

A scanning electron microscope Quanta 200 FEG (FEI, The Netherlands) was used to observe blend morphologies. For these characterizations, the samples were cryofractured under liquid nitrogen, either in the transverse or in the parallel direction of the thread collected after extrusion, and fracture surfaces were coated with carbon. The transmission mode (STEM) was also used. In this case, samples were ultramicrotomed under liquid nitrogen with a Leica apparatus, EM UC7 (Germany). All STEM micrographs were recorded at an accelerating voltage of 10 kV.

Cryo-ultramicrotomy with the EM UC7 apparatus was also performed on threads in the transverse direction, in order to prepare sample surfaces for AFM characterizations. PLA80Si5 and PLA80SiR5 blends were then tested with the MFP-3D Infinity (Asylum Research). A bimodal tapping mode (AMFM) was set, allowing to obtain mapping of surface topography, phase, and stiffness. A silicon probe (AC160R3) was used with a spring constant of 26 N/m and a resonant frequency of 300 kHz. A scan rate of 1 Hz was set for each test.

For each composition, 900 mg of extruded thread were immersed into 15 mL of chloroform at room temperature and stirred under ultrasonic probe during 48 h to dissolve the PLA. Then, PA11 nodules were purified by three washing/centrifugation cycles (10,000 rpm, 5 min) using chloroform and finally collected for analysis. A Coulter LS 13230 (Coulter Beckmann Co., USA) laser diffraction particle size analyzer instrument was used to determine the size distribution of extracted PA11 nodules. Size measurements were performed using the micro liquid module (15 mL) in chloroform. Obscuration was 10 ± 2%. Three measurements were performed for each sample. Laser diffraction particle size analyzer is an interesting alternative method to characterize dispersed phases in immiscible polymer blends. In fact, the number of dispersed phases analyzed by this method is much larger than the number analyzed with conventional image analysis from electron microscopy observations (SEM or TEM).

### 2.6. Interfacial Tension and Wetting Parameter

To predict the final localization of silica in the polymer blend nanocomposites, while considering thermodynamic parameters at room temperature, the wetting coefficient *ω_AB_* was calculated based on Equation (1) [6].
(1)$$\omega_{AB}=\frac{\gamma_{SB}-\gamma_{SA}}{\gamma_{AB}}$$
where *γ_SB_*, *γ_SA_*, and *γ_AB_* are the interfacial tensions between silica and polymer B, silica, and polymer A and both polymers A and B, respectively.

Interfacial tensions were obtained from the mean harmonic and geometric equations of Wu for polymer/polymer and polymer/silica interfacial tensions, respectively as shown in Equations (2) and (3) [19]:

Mean harmonic equation of Wu for *γ_AB_*
(2)$$\gamma_{ij}=\gamma_{i}+\gamma_{j}-4\left(\frac{\gamma_{i}^{d}\gamma_{j}^{d}}{\gamma_{i}^{d\;}+\gamma_{j}^{d}}+\frac{\gamma_{i}^{p}\gamma_{j}^{p}}{\gamma_{i}^{p}+\gamma_{j}^{p}}\right)$$

Mean geometric equation of Wu for *γ_SB_* and *γ_SA_*
(3)$$\gamma_{ij}=\gamma_{i}+\gamma_{j}-2\left(\sqrt{\gamma_{i}^{d}\gamma_{j}^{d}}+\sqrt{\gamma_{i}^{p}\gamma_{j}^{p}}\right)$$
with $\gamma_{ij},$ the interfacial tension between components i and j (i and j can be a polymer or silica), $\gamma_{i}$ the surface tension of component *i* and $\gamma_{i}^{d}$ and $\gamma_{i}^{p}$ the dispersive and polar contributions of the surface tension of the same component, respectively.

Dispersive and polar contributions of the surface tension for each component (PLA, PA11, silica A200, and silica R805) were obtained by contact angle measurements using distilled water and diiodomethane as liquids deposited on injection molded (for polymers) or compression molded (for silica, performed in another article [20]) disks of each component. The Owens–Wendt equation was used to calculate $\gamma_{i}^{d}$ and $\gamma_{i}^{p}$ from the contact angle θ (Equation (4)).
(4)$$\gamma_{L}=\left(1-\cos\theta \right)=2\;\sqrt{\gamma_{S}^{d}\gamma_{L}^{d}}+2\sqrt{\gamma_{S}^{p}\gamma_{L}^{p}}$$

The measurement of the contact angle θ between the liquid and the disk was performed with a goniometer DSA30 Series apparatus (Krüss, Germany) equipped with the KRUSS-AVANCE 1.5.1.0 software.

The methodology of the results part (below) consists in proving first that the amount of silica of all formulations was effectively 5 wt %. This is necessary to compare rheological tests performed on each formulation. Secondly, the oscillatory shear rheology was conducted on neat polymer nanocomposites to investigate the dispersion (and affinity) of each silica in each polymer. Then, polymer blend nanocomposites were investigated. Indeed, their rheological behavior and their microstructure were studied. Finally, the localization of both silica was compared to that obtained by calculating the wetting parameter. Then, all those results were discussed based on the results in the literature.

## 3. Results

### 3.1. Thermogravimetric Analysis

The thermogravimetric analysis (Figure 1 and Appendix A) allows to evidence first that the R805 modified Aerosil silica exhibits a 5.5 wt % loss at 650 °C compared to the A200 that exhibits only a 0.3 wt % loss. This is due to the decomposition of trimethoxyoctylsilane grafted at the surface of R805 silica. The second observation is that both PA11 formulations containing silica have a higher stability than PLA ones due to the higher thermal stability of PA11 compared to PLA. Moreover, the derivative thermal gravimetry (DTG) peak for PA11Si5 is reached at 460 °C whereas it is at 474 °C for PA11SiR5. Considering a peak of DTG for neat PA11 at 505 °C (not shown here), both silica incorporations in the PA11 lead to a decrease of the thermal stability whereas this decrease is less pronounced for R805. The DTG peak of PLASi5 and PLASiR5 is reached at 364 °C. Here again the incorporation of either A200 or R805 silica decreases the thermal stability of the PLA. The last and most important information provided by Table 4 is the real amount of silica incorporated in each polymer. It can be concluded that the four formulations exhibit a final amount of silica very close to the 5 wt % targeted.

The thermogravimetric measurements performed on the blends (Figure 2) evidenced first that PLA80 has a thermal behavior close to the linear rule of mixture (LRM). Moreover, the filled blends PLA80Si5 and PLA80SiR5 exhibit a slightly better thermal stability than the neat PLA80 blend, all along the temperature range. Both exhibit almost the same thermal degradation profile, with however a small difference favorable to PLA80Si5. The amount of silica in both filled blends is almost the same (Table 4).

### 3.2. Oscillatory Shear Rheology on Neat Polymer Nanocomposites

Rheological tests were conducted in order to evaluate the complex viscosity of each formulation. A second interest of rheological measurements is to understand more about the final microstructure of each formulation. Finally, interactions between the different components in each formulation can be highlighted by this kind of tests. Regarding the strain sweep test, while pure polymers exhibit a plateau up to 50% of strain (not shown here), the strain sweep curves of pure polymers filled with both silica start to decrease at a lower strain than 50%. Moreover, their behaviors are quite different (Figure 3). The linear zone goes up to 7% strain for PLASi5, PLASiR5, and PA11Si5 whereas it goes only up to 2% strain for PA11SiR5. This is a first proof of the formation of a solid network in the case of PA11SiR5. A high reinforcement efficiency of silica R805 in PA11 is seen in Figure 3.

Considering a zero shear viscosity of 95 and 330 Pa.s at 210 °C for PLA and PA11, respectively, the complex viscosity of each polymer filled with silica exhibits a high increase all along the frequency range compared to the pure polymers (Figure 4).

G′ ∝ ω^2^ with G″ ∝ ω (in double logarithmic plot) is typical of a viscoelastic liquid (terminal Maxwellian behavior) whereas G′ ∝ ω with G″ ∝ ω is typical of a viscoelastic solid. This last behavior is classically observed at low frequency for polymeric materials filled with micro- or nanoparticles. Table 5 summarizes the slope calculated from G′ and G″ versus ω for each polymer with either A200 or R805 silica.

It is clear that PA11SiR5 and PLASi5 tend to form a viscoelastic solid as the slope of G′ vs. ω gets close to 1.

The dispersion of R805 in PA11 and A200 in PLA leads to a solid network with a probable percolation of the fumed silica.

The Carreau Yasuda model is usually used to qualify the state of dispersion of a minor phase into a composite [21,22]. In this model, the complex viscosity $\eta^{*}\;$ is non-linearly related to the frequencies ω through five independent variables
(5)$$\eta^{*}\left(\omega \right)=\frac{\sigma_{0}}{\omega }+\eta_{0}\;{\left[1+{\left(\lambda \omega \right)}^{a}\right]}^{\frac{n-1}{a}}$$
with $\sigma_{0}$ the yield stress, $\eta_{0}$ the viscosity at zero shear rate, λ a characteristic time (relaxation), ω the frequency, a the Yasuda parameter, and n the flow index.

Microsoft Solver© in Excel© was used to obtain the best fit with the experimental data of Figure 4. The yield stress is used to be related to minor phase’s dispersion (in the present case silica nanoparticles dispersion): the greater the value is, the better dispersed is the minor phase into the composite. The nanocomposites for which the silica has the best dispersion is PA11 with R805 followed by the PLA with A200 (Table 6). The PLA with A200 exhibits the lower chain mobility (with the higher relaxation time). The flow index is nearly the same for all formulations.

### 3.3. Oscillatory Shear Rheology on Polymer Blend Nanocomposites

The viscoelastic behavior of the PLA80 blend (Figure 5) is typical of matrix/dispersed phase blends with a complex viscosity near that of the PLA matrix at high frequency and in-between PLA and PA11 at medium frequency. Moreover, at low frequency, G′ of the blend is getting higher than that of each neat polymer. This is due to the relaxation of the PA11 dispersed phase and is called shape relaxation behavior. It must be noticed here that it was not possible to measure G′ at lower frequency than 0.2 rad/s, as the value is too low to be collected by the machine.

Table 7 indicates the range of the linear viscoelastic behavior for the neat blend and blend nanocomposites. The linear zone is narrowed for both PLA80Si5 and PLA80SiR5. In the case of PLA80SiR5 blend, the linear zone is shorter just as in the case of PA11SiR5. Then, it is probable that a strong solid network is formed for this PLA80SiR5 blend.

Frequency sweep tests (Figure 6) confirm the formation of a solid network as the complex viscosity dramatically increases at low frequency when adding 5 wt % of silica, leading to a yield behavior of both PLA80Si5 and PLA80SiR5. G′ and G″ vs. ω are in the Appendix A.

Comparing Figure 6 with Figure 4, it is clear that the yield behavior is much more pronounced for blend nanocomposites than neat polymer nanocomposites. It means that adding a dispersed PA11 polymeric phase into PLA dramatically change the rheology and therefore the microstructure of the nanocomposite. This change in the microstructure is directly linked to the state of dispersion and localization of the fumed silica in the blend. That will be discussed later.

### 3.4. SEM Microstructure

SEM images of the blend nanocomposites were compared to that of the neat blend. Figure 7 shows the microstructure of those three samples perpendicularly to the direction of the thread collected after extrusion. The shape and size of the PA11 dispersed phases seem to vary depending on the type of silica. PA11 dispersed nodules are very big in PLA80Si5 whereas they look small and spherical for PLA80SiR5. Moreover, some roughness can be seen at the surface of PA11 nodules in the case of PLA80SiR5. This roughness can be due to the R805 silica particles localized at the surface of the nodules.

The microstructures of the same three samples cut parallel to the direction of the thread (Figure 8) clearly show no spherical PA11 dispersed phases for PLA80Si5 and PLA80SiR5. Figure 8 evidences that PA11 nodules are small and almost spherical in the case of PLA80SiR5 whereas they are big and elongated in the case of PLA80Si5.

### 3.5. PA11 Dispersed Phase Size

To investigate more about the size and shape of the PA11 dispersed phase for PLA80, PLA80Si5, and PLA80SiR5, determination of the particle size distribution of the PA11 dispersed phases using laser diffraction was performed (Figure 9). PA11 phases exhibit a spherical shape in PLA80 blend with a d_v_ = 1.82 µm; whereas it is clear that for PLA80Si5 and PLA80SiR5, PA11 phases are not spherical, as the curves exhibit different peaks. Table 8 shows a large increase in PA11 diameters for blend nanocomposites compared to the neat blend with a huge increase for A200 filled blend (PLA80Si5).

To observe the shape of PA11 dispersed phase, the samples were collected after extraction of PLA with chloroform and observed under SEM. Figure 10 shows the PA11 dispersed phases after extraction (by dissolution) of the PLA matrix and after size distribution measurement. In the case of the neat blend, spherical nodules are seen. For PLA80Si5, the shape of the PA11 dispersed phase is clearly not spherical but elongated. These two microstructures are in good agreement with the PA11 size distribution shown in Figure 9. Finally, in the case of PLA80SiR5, the PA11 nodules are spherical with R805 silica at the surface of the nodules. By comparing Figure 9 and Figure 10, it can be assumed that attractive interactions exist between the PA11 nodules leading to a large distribution of the size of the PA11 dispersed phase.

### 3.6. Wetting Parameter Calculation

Regarding Table 9, the measured values of surface tension (in mN/m) for each component are in good agreement with the literature for PLA and PA11 [15]. Regarding the localization of the silica NPs in the blend, the wetting parameter (Table 9) predicts that R805 should be localized at the interface and A200 in the PLA phase. On the other hand, STEM micrographs of Figure 11 show that silica A200 nanoparticles are localized in the PA11 nodules, whereas silica R805 NPs are mainly localized close to the surface of the PA11 nodules. This was also shown by SEM images (Figure 7, Figure 8 and Figure 10). Hence, the real localization fits well with the thermodynamic prediction in the case of R805 silica whereas it is not the case for A200 silica.

Firstly, regarding the shape and size of the PA11 dispersed phase, AFM images ((Figure 12) correlate with the particle size analysis (Figure 9). Indeed, PA11 dispersed nodules are larger and less spherical in the case of PLA80Si5. Moreover, AFM images (Figure 12) correlate with the SEM (Figure 10) and STEM (Figure 11) micrographs, showing A200 silica uniformly dispersed in the PA11 dispersed phase. In the case of PLA80SiR5, AFM images clearly show that there are R805 silica NPs at the interface. Moreover, topography and phase images for PLA80SiR5 clearly evidence two kinds of ‘particles’ pointed with black and white arrows on the phase image, whereas stiffness images of PLA80SiR5 do not show any differences. It seems that the localization of R805 silica at the interface sticks some PA11 dispersed droplets together leading to clusters of droplets. White arrows point the silica fully covered PA11 droplets, whereas black arrows point the partially covered PA11 droplets. This will be discussed later.

## 4. Discussion

The viscosity of the polymeric phases such as the ratio of viscosity ($p=\frac{\eta_{d}}{\eta_{m}}$, where *η_d_* is the viscosity of the PA11 dispersed phase and *η_m_* is the one of the PLA matrix) are known to play a key role in the final dispersion of nanoparticles. The literature remains controversial regarding the influence of those parameters. Three different works are described below to highlight that the influence of viscosity and viscosity ratio is not yet fully understood. Feng et al. [23] studied the localization of carbon black nanoparticles (CB NPs) either in PMMA dispersed phase or PP matrix depending on the PMMA viscosity. They observed that CB NPs were dispersed into the PMMA preferred phase when the viscosities of PMMA and PP are comparable, whereas by increasing the viscosity of PMMA, CB NPs tend to disperse at the interface and in the PP phase. In that case, they concluded that the high viscosity of PMMA droplets inhibits the diffusion of CB NPs inside them. The process is known to play a crucial role in the final morphology of the blend and especially in the localization of the NPs. The authors mentioned that the blend was melt mixed with an internal mixer at 190 °C and 30 rpm without any more details regarding the processing route [23]. In a more recent work, Plattier et al. [24] studied the localization of CB fillers (particle size in the range of 200–400 nm) into a co-continuous PP/PCL blend regarding the viscosity ratio. As processing, the authors first melt mixed the blend (PP and PCL) by a microcompounder and then added the CB fillers. CB fillers are known to prefer the PCL phase. The authors observed that CB fillers were systematically dispersed in the most viscous phase except for the viscosity ratio of 1 for which they were dispersed at the interface. The authors explained their results in terms of hydrodynamic forces acting on the CB fillers. Fillers are extracted to the most viscous phase that applies the most important forces. When the viscosity ratio is close to 1, the two forces balance each other and the CB fillers are localized in between the two phases [24]. Finally, Favis et al. published a series of works dealing with the localization of different particles (nanosilica, microsilica, and nanowires of copper) into two different matrix/dispersed phase blends: low interfacial tension one (PLA/ Polybutylene adipate terephthalate, PBAT) and high interfacial tension one (PLA/Low density polyethylene, LDPE) [25,26,27,28]. The viscosity ratio for the PLA/LDPE and PLA/PBAT blends is 0.83 and 0.12, respectively. Hence, we can consider a viscosity ratio lower than 1. The two polymeric phases were first added into the internal mixer. Then the fillers (mentioned previously) were incorporated in a second step. Only the conclusions regarding the process in which all components were added together are presented here. Whatever the interfacial tension (low or high) and whatever the viscosity ratio between the two polymers, the fillers were always dispersed into the most preferred phase: i.e., PLA most viscous matrix phase for PLA/LDPE and PBAT less viscous dispersed phase in the case of PLA/PBAT. This illustrates that whatever the viscosity ratio, interfacial tension, composition of the blend and aspect ratio of the particles, the final localization is the one predicted by the wetting parameter. Those different works highly illustrate the difficulty to converge into one main conclusion regarding the influence of the viscosity and the viscosity ratio of polymeric phases on the final localization of a (nano)particle.

In our case, for silica R805, thermodynamic prediction correlates with the observation as this silica goes to the interface. It is also in good agreement with Favis et al. observation regarding the transfer to the interface of nanosilica, while using a two steps process and low interfacial tension blend (PLA/PBAT) [27,28]. For silica A200, the conclusions are different as the wetting parameter predicts that those silica NPs should localize in the PLA matrix whereas they are mainly localized in the PA11 dispersed phase. As PA11 is more viscous than PLA ($p=\frac{\eta_{d}}{\eta_{m}}=3.4$), it can be assumed that, during the process, silica NPs are extracted from the PLA phase toward the highest stress-applying PA11 phase. Moreover, this extraction is accompanied by an elongation of the PA11 dispersed phases. Hence, the elongated shape of PA11 phases observed in the case of PLA80Si5 can be explained by the increased yield stress of the filled PA11 phase, accompanied by a slowdown of the relaxation. This stabilizes the elongated morphology leading to a quasi-co-continuous morphology. We propose to discuss the obtaining of a quasi-co-continuous morphology with A200 by comparing what is seen in the literature.

This phenomenon was described by Liu et al. [13], Pawar et al. [8] and Wu et al. [14] and is due to the NPs that drag the domains of wetting polymer causing their co-continuity formation. Indeed, Wu et al. have stated the self-networking of 10 phr of carbon black fillers into ABS/PA6 (80/20) blends leading to co-continuity at such low amount of PA6 [14]. In their work, they showed a SEM micrograph of the ABS/PA6 (80/20) filled with 15 phr of CB after selective extraction of ABS by tetrahydrofuran, THF. Their microstructure is very close to that of Figure 10 for PLA80Si5. It is obvious that A200 silica nanoparticles entrapped in a highly elongated PA11 phase will lead to the formation of a solid network identified under rheological tests at low frequency [29]. Finally, the localization of A200 in the blend, the shape of PA11 phases as well as the rheological behavior of PLA80Si5 is now well-understood. What about the microstructure and rheology of PLA80SiR5?

R805 silica is shown to segregate at the interface of our PLA/PA11 blend. Usually, elongation of the dispersed phase is expected when nanosilica find their way to the interface during processing. This is clearly described by Jalali et al. [27], who observed and explained the formation of an elongated PBAT dispersed phase into 70/30 PLA/PBAT blend filled with 3 wt % of nanosilica. In their case η*_PBAT_ < η*_PLA_, whereas in our case PA11 dispersed phase have a higher viscosity than PLA matrix. This parameter can explain on his own the difference of dispersed phase shape observed. Hence, in our PLA80SiR5, silica NPs are localized at the interface with nearly spherical shape of the dispersed PA11 phase [27]. Now, how to explain their rheological behavior, and especially the gel-like behavior evidenced at low frequency during frequency sweep test? Here again, we propose to discuss the rheological behavior of our polymer blends nanocomposites by comparing what is seen in the literature.

Velankar et al. [30] were among the first to describe the particle-bridged drop cluster phenomenon in which particles glue two immiscible polymers. Zou et al. [31] has taken up this theory and described the microstructure and rheological behavior obtained for 90/10 and 10/90 polybudadiene/polydimethylsiloxane (PBD/PDMS) blends filled with various amounts of hydrophobic fumed silica. The silica are supposed to go to the interface. In their case, as polymers are in the liquid form at room temperature, the process was very simple: hand-mixing with a spatula in a small Petri dish until a homogeneous mixture was formed. At high loadings (>1 wt %), when the droplet surface was completely covered with particles, as fumed silica exhibit high aspect ratio, fractal-like shape and high interparticle attractions (i.e., O-Si strong ionic bonds), they have a high tendency to flocculate. In their blend, this gives a peculiar morphology of the system described as droplet clusters structure [31].

Those previous results allow highlighting the microstructure and rheology of our PLA80SiR5. In our case, as shown by AFM, the surface of some PA11 dispersed phase is fully covered by fumed silica, whereas other PA11 droplets are only partially covered by silica. Moreover, as fumed silica has a fractal structure, it can flocculate and one floc can belong to different PA11 nodules. With a strong iono-covalent link between the nanometric individual particles, this leads to a strong link between each PA11 spherical nodules, that look stuck together. This can explain the solid network visible at low frequency under rheology. Both final microstructures are schematically represented in Figure 13.

## 5. Conclusions

A200 and R805 silica form solid networks in PLA and PA11 respectively, as shown by frequency sweep tests. This rheological behavior is extended to the 80/20 PLA/PA11 blend, as both A200 and R805 silica form a solid network (as shown by the yield behavior of the blend nanocomposites). For the PLA80Si5 blend nanocomposite, the localization of A200 in the most viscous PA11 dispersed phase can be explained by an extraction from PLA phase toward the most viscous phase that apply the most important hydrodynamic forces. This is accompanied by an elongation of the PA11 droplets leading to a quasi-co-continuous morphology of the blend. For the PLA80SiR5 blend nanocomposite, silica NPs are localized at the interface, as predicted by thermodynamics and the yield behavior identified by frequency sweep tests in rheology is due to the fractal nature of those silica NPs that create bridges between the dispersed droplets. These final microstructures correlate with the literature.

Peculiar morphologies are then obtained with a 80/20 bio-based polymer blend filled with hydrophilic or hydrophobic fumed silica. Being able to achieve co-continuous morphologies at very low levels of dispersed phase opens up possibilities. In addition, the dispersed phase bridges created by fumed silica could improve other properties (mechanical such as stiffness, impact, toughness). Finally, by replacing the fumed silica by electrically conductive nanoparticles, it can be expected to have a percolated network and a high electrical conductivity.

## Figures and Tables

**Figure 1 nanomaterials-11-01721-f001:**
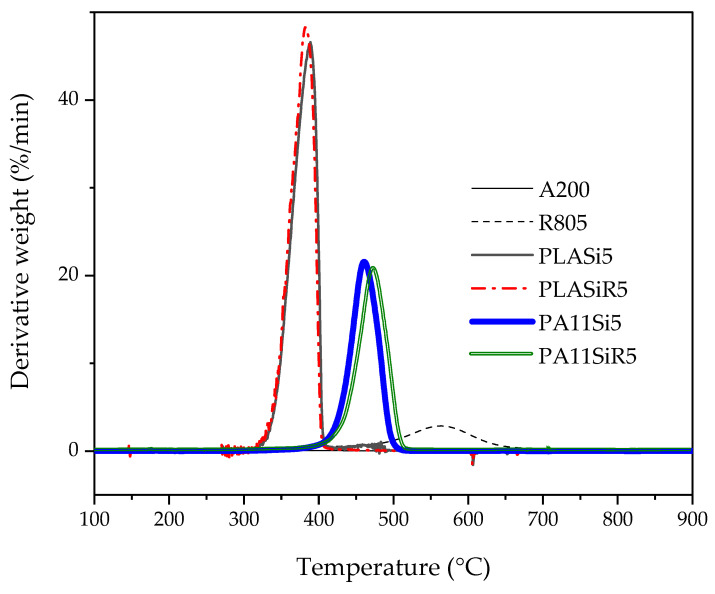
DTG of pure silica (A200 and R805) and of neat polymer nanocomposites (PLA and PA11 with each silica A200 and R805). It must be noted that the DTG peak of neat PA11 is at 505 °C whereas that of neat PLA is at 403 °C.

**Figure 2 nanomaterials-11-01721-f002:**
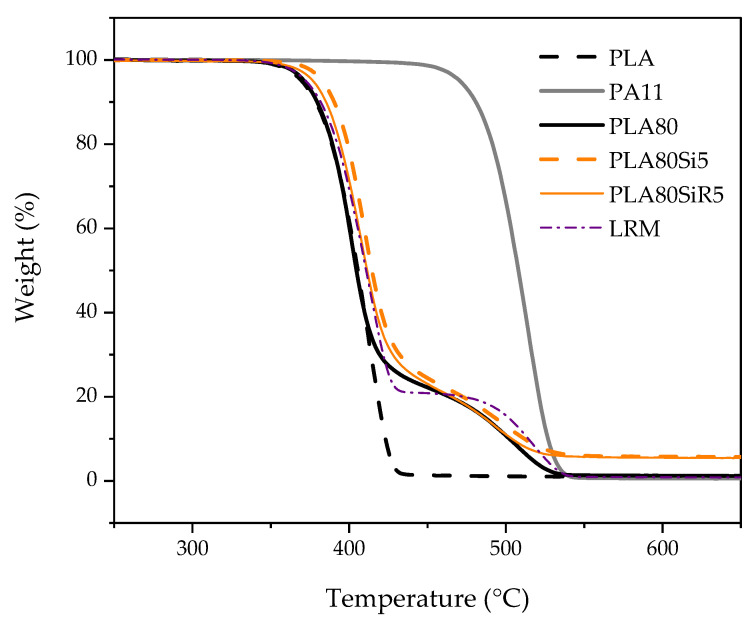
TGA measurements of pure PLA, PA11, PLA80 blend and polymer blend nanocomposites (PLA80Si5 and PLA80SiR5). The PLA80 thermogram is compared to that of the linear rule of mixture (LRM).

**Figure 3 nanomaterials-11-01721-f003:**
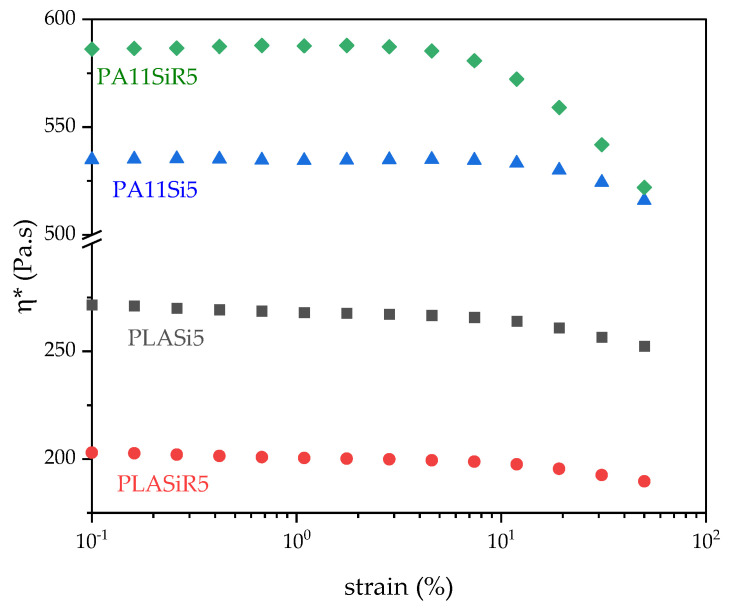
Complex viscosity versus strain (%) for the neat polymer nanocomposites.

**Figure 4 nanomaterials-11-01721-f004:**
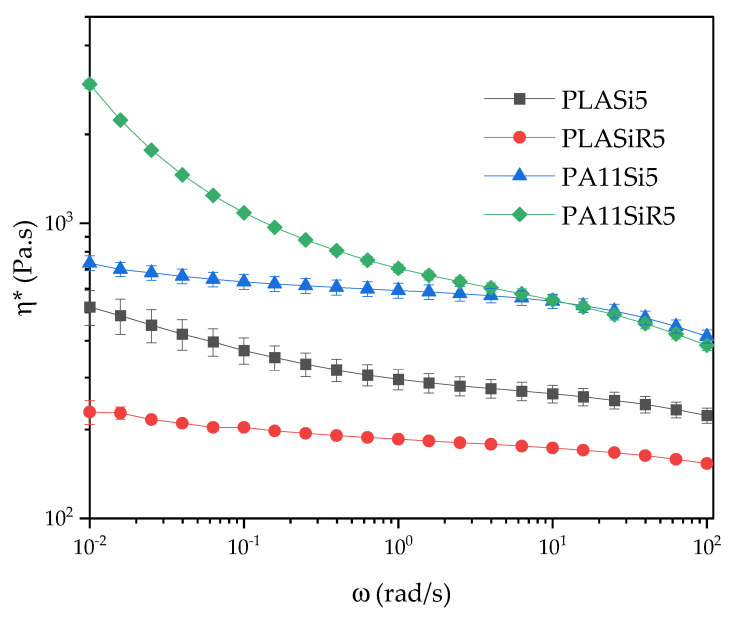
Complex viscosity η*, G′, and G″ versus frequency for neat polymer nanocomposites.

**Figure 5 nanomaterials-11-01721-f005:**
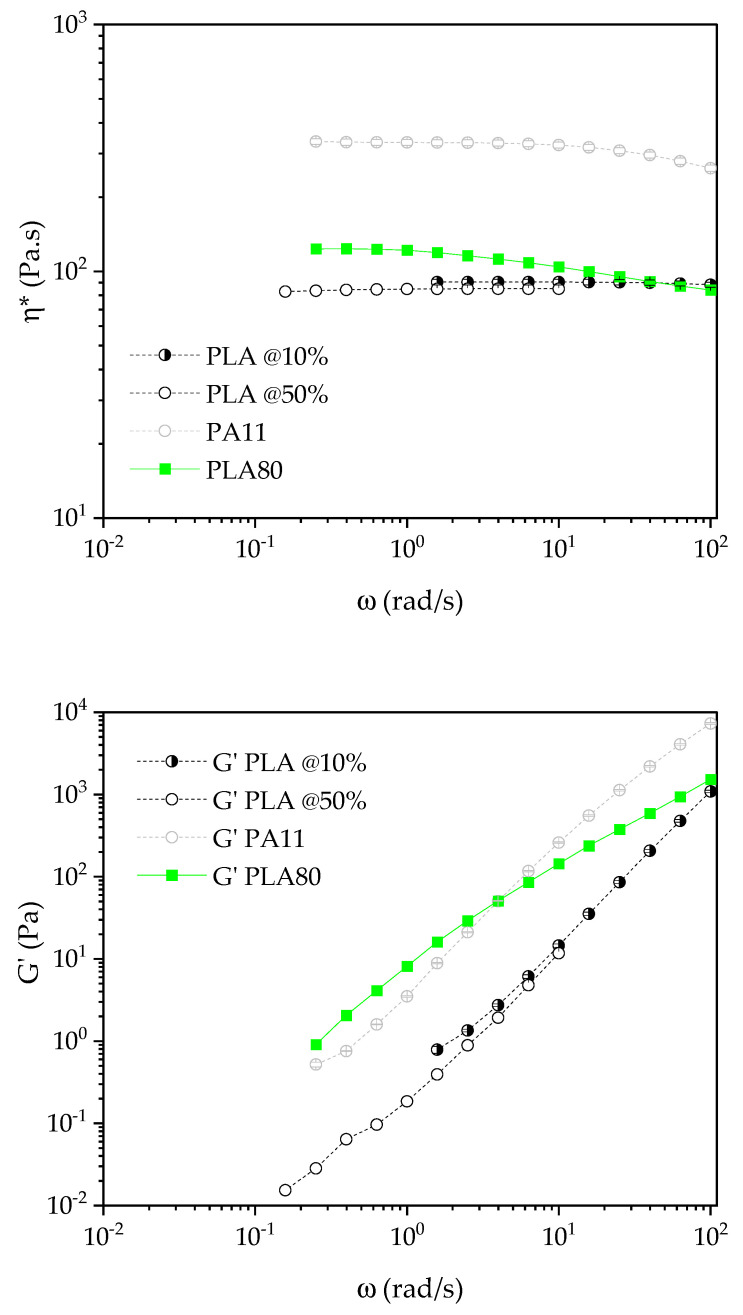
η* and G′ vs. ω for the neat polymer and the PLA80 blend. Measurements on PLA were performed at strains of 10% and 50%.

**Figure 6 nanomaterials-11-01721-f006:**
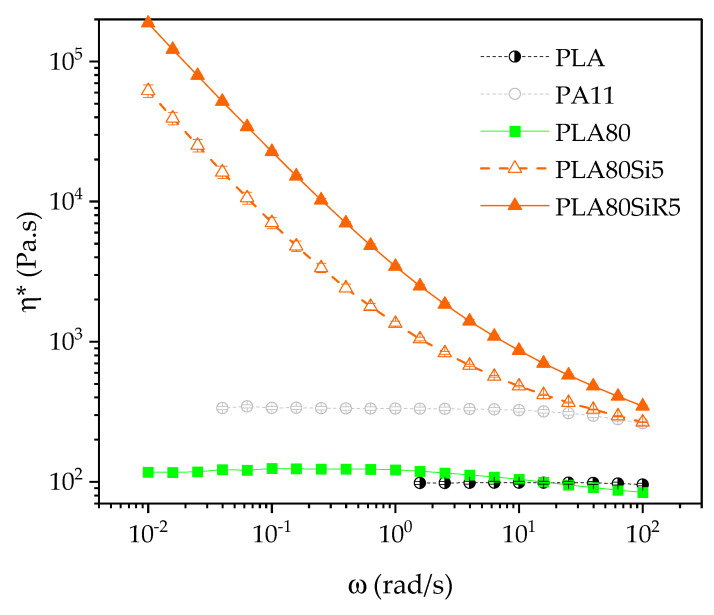
Frequency sweep tests giving the complex viscosity versus frequency for the neat polymers, PLA80 blend and both blend nanocomposites PLA80Si5 and PLA80SiR5.

**Figure 7 nanomaterials-11-01721-f007:**
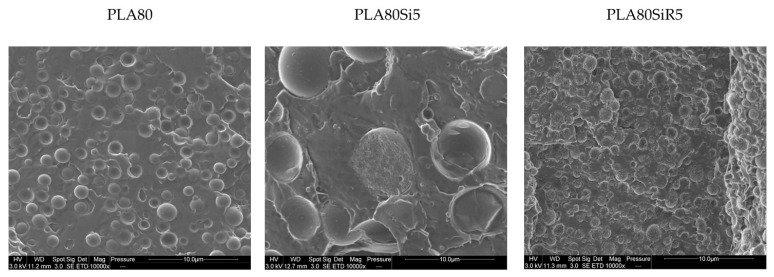
SEM micrographs of threads after extrusion for the PLA80, PLA80Si5, and PLA80SiR5. The samples were cut perpendicular to the direction of the thread.

**Figure 8 nanomaterials-11-01721-f008:**
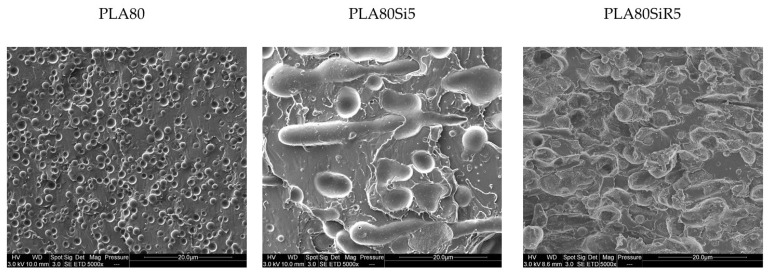
SEM micrographs of threads after extrusion for the PLA80, PLA80Si5, and PLA80SiR5. The samples were cut parallel to the direction of the thread.

**Figure 9 nanomaterials-11-01721-f009:**
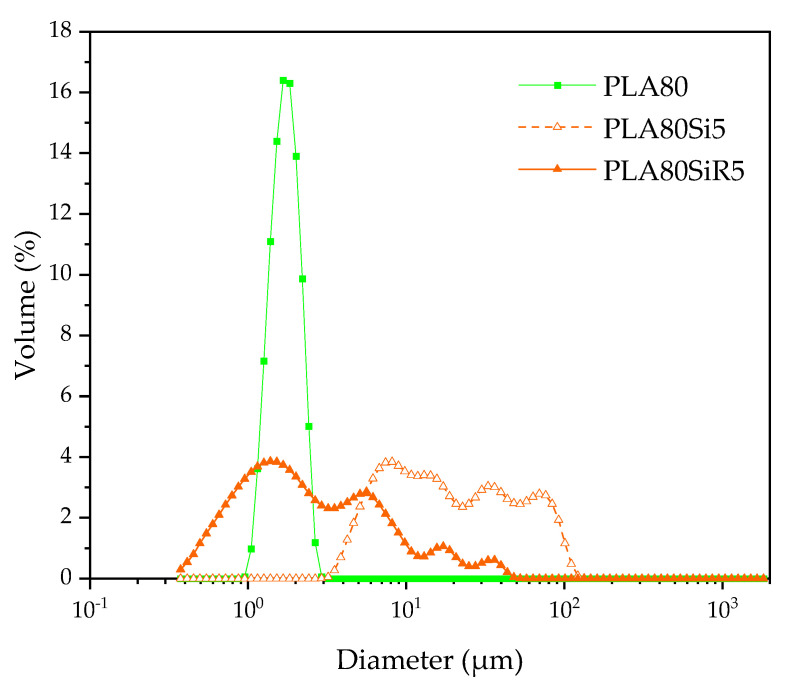
Volume diameter of the PA11 dispersed phases obtained by laser diffraction in chloroform.

**Figure 10 nanomaterials-11-01721-f010:**
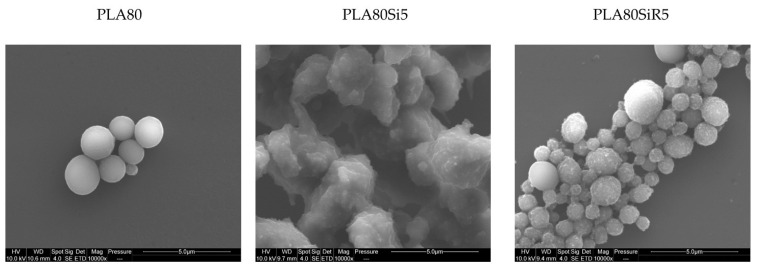
SEM micrographs of the PA11 nodules after extraction of PLA with chloroform for PLA80, PLA80Si5, and PLA80SiR5.

**Figure 11 nanomaterials-11-01721-f011:**
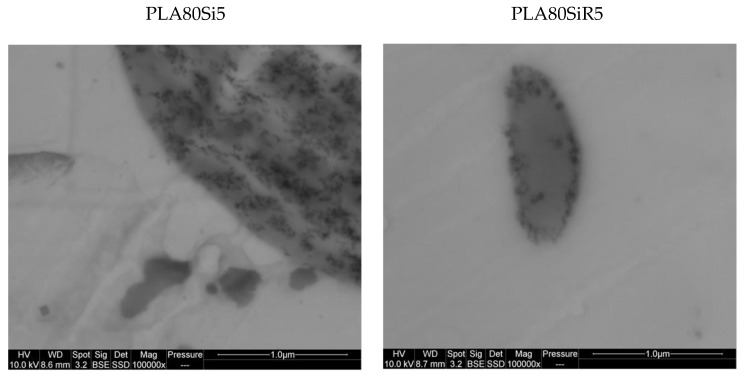
STEM micrographs of PLA80Si5 and PLA80SiR5.

**Figure 12 nanomaterials-11-01721-f012:**
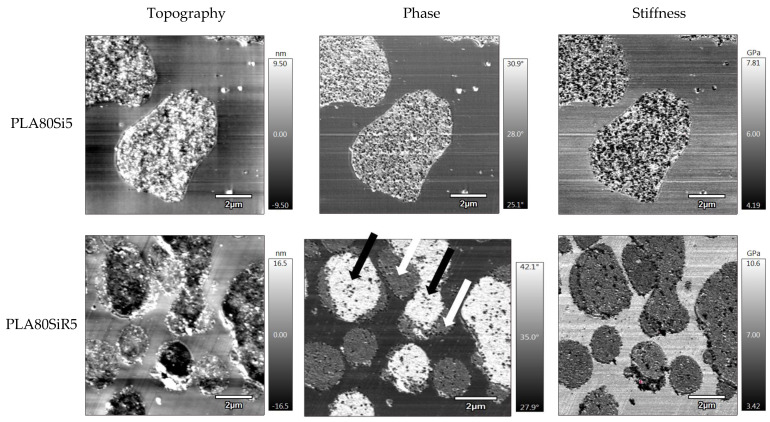
Topographic, phase, and stiffness AFM images for PLA80Si5 and PLA80SiR5.

**Figure 13 nanomaterials-11-01721-f013:**
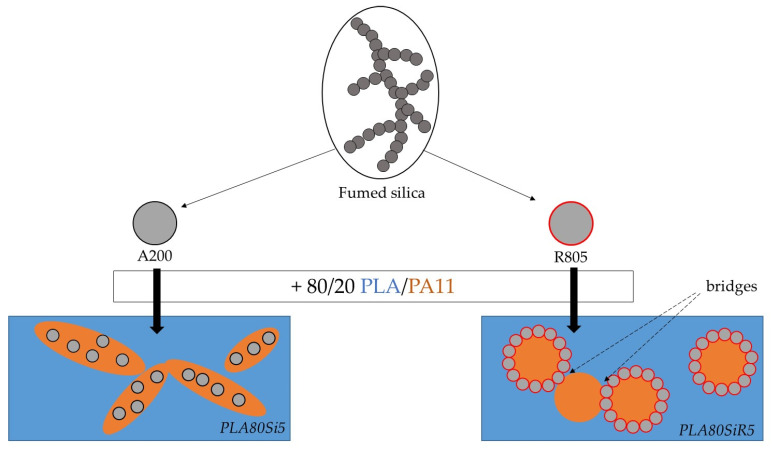
Schematic representation of the final microstructures of PLA80Si5 and PLA80SiR5 nanocomposites.

**Table 1 nanomaterials-11-01721-t001:** Characteristics of Aerosil silica.

	Aerosil A200	Aerosil R805
Carbon wt %	~0	4.5–6.5
Specific surface area (m^2^/g)	200 ± 25	150 ± 25
Average diameter of primary particles (nm)	12	12

**Table 2 nanomaterials-11-01721-t002:** Extrusion conditions.

	Screw Speed (rpm)	Flow (kg/h)	Feed Zone Temperature (°C)	Temperature of the Other 12 Zones (°C)
Nanocomposites PLA/silica and PA11/silica	250	6	60	180 for PLA 210 for PA11
Blends of PLA80, PLA80Si5, and PLA80SiR5	200	4	80	210

**Table 3 nanomaterials-11-01721-t003:** Weight % of each component in each formulation.

	PLA	PA11	A200	R805
PLASi5	95	0	5	0
PLASiR5	95	0	0	5
PA11Si5	0	95	5	0
PA11SiR5	0	95	0	5
PLA80	80	20	0	0
PLA80Si5	76	19	5	0
PLA80SiR5	76	19	0	5

**Table 4 nanomaterials-11-01721-t004:** Real amounts of silica in each polymer (measured at 650 °C).

	PLA Si5	PLA SiR5	PA11 Si5	PA11 SiR5	PLA80Si5	PLA80SiR5
Real amount (wt %) of silica	4.45 ± 0.2	5.09 ± 0.08	5.05 ± 0.4	4.85 ± 0.07	5.69 ± 1.3	5.75 ± 0.13

**Table 5 nanomaterials-11-01721-t005:** Slopes of the curve G′ and G″ versus ω (in log-log) at low frequencies (0.01 to 1 rad/s).

Samples	Coefficient x in G′ ∝ ω^x^	*R* ^2^	Coefficient x in G″ ∝ ω^x^	*R* ^2^
PLA	1.88	0.9998	0.99	1
PLA Si5	1.08	0.9999	0.94	0.9890
PLA SiR5	1.22	0.9999	0.96	0.9955
PA11	1.69	0.9969	0.97	0.9998
PA11 Si5	1.39	0.9991	0.96	0.9999
PA11 SiR5	1.03	0.9995	0.87	0.9973

**Table 6 nanomaterials-11-01721-t006:** Parameters obtained from Carreau Yasuda model in Equation (5).

	$\sigma_{0}$ (Pa)	$\eta_{0}$ (Pa.s)	λ (s)	a	n	*R* ^2^
PLASi5	0.2	520	2049.8	53.2	0.75	0.99
PLASiR5	0.0	220	116.0	53.3	1.0	0.99
PA11Si5	0.0	733	134.8	53.2	1.0	0.97
PA11SiR5	23.8	2780	92.0	70.0	0.9	0.99

**Table 7 nanomaterials-11-01721-t007:** Zone of viscoelastic linear behavior.

Samples	Deformation (%)
PLA80	Up to 20%
PLA80Si5	Up to 4%
PLA80SiR5	Up to 1%

**Table 8 nanomaterials-11-01721-t008:** PA11 dispersed phase size obtained by laser diffraction.

	Median Diameter in Volume d_m_ (µm)	Mean Diameter in Volume d_v_ (µm)	Mean Diameter in Number d_n_ (µm)
PLA80	1.80	1.82	1.61
PLA80Si5	18.27	29.79	6.56
PLA80SiR5	2.28	4.99	0.75

**Table 9 nanomaterials-11-01721-t009:** Interfacial tension measured via contact angle measurement and calculated via Owens Wendt and wetting parameter calculated via Young equation.

	$\gamma^{d}$ (mN/m)	$\gamma^{p}$ (mN/m)	γ (mN/m)	ω
PLA 3251D	33.8	5.0	38.7	
PA11 LMFO	37.4	4.6	42	
Aerosil A200 ^1^	29.4	50.6	80.0	0.63
Aerosil R805 ^1^	33.5	4.1	37.6	−1.04

^1^ From [20].

## Data Availability

Not applicable.

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
