# Peer review of "Peculiar Morphologies Obtained for 80/20 PLA/PA11 Blend with Small Amounts of Fumed Silica"

_nanomaterials, 2021, doi:10.3390/nano11071721_

Round 1
Reviewer 1 Report
I think paper is very interesting and I have only one suggestion reported in file attached

Author Response
The authors want to thank the reviewers for their critical comments and constructive suggestions. Based on these comments and suggestions, we have revised the original manuscript and hope that these revisions improve the quality of the manuscript and the reviewers consider it worthy for publication in Nanomaterials.
Reviewer 1 suggests that TGA curve should be added to Figure 1 “I think it is better to show also TGA curves not only DTGA in paper rather than in supplementary materials”.
DTGA is plotted in Figure 1 whereas, the TGA curves are plotted in Figure S1 in the Supporting Information. We understand reviewer’s comment but we think it is better to keep only DTGA in the manuscript and add TGA in supporting information in order not to weight down Figure 1.

Reviewer 2 Report
This study “Peculiar morphologies obtained for 80/20 PLA/PA11 blend with small amounts of fumed silica” focuses on the effect of spherical silica nanoparticles 87 bearing a fractal structure fumed silica on the final morphology of a 80/20 PLA/PA11 blend.
-Authors should describe the novelty of the work and the difference between this work with other similar research.
- The language used in the manuscript can be more specific to the scope and aim of the study.
- The methodology section is not well organized for the readers to understand the concept.
- The conclusion section is very general. Must be rewritten and revised.
- What’s authors target from lines 471-511 in the discussion? did they explain the mechanism of this work?
Page 20 “. However, the authors only mentioned that the blend was melt mixed with an internal mixer at 190°C and 30rpm without any more details regarding the processing route (sequence of mixing, especially to avoid particles incorporation in the phase that melts first (i.e. PP)).” These sentences do not make any sense at all. Please revise for improved clarity in order to better understand the readership.
- The authors should review the other investigation on their study way in the introduction part and finally note the novelty of the article. The introduction part needs to develop. Authors can use below references and related works in the field:
High-performance fully bio-based poly (lactic acid)/polyamide11 (PLA/PA11) blends by reactive blending with multi-functionalized epoxy. Polymer Testing, 2019, 78. 105980.
Evaluation of clay and fumed silica nanoparticles on adsorption of surfactant polymer during enhanced oil recovery. Journal of the Japan Petroleum Institute, 2017, 60. 85-94.
- The authors should indicate experimental errors throughout the paper.
- Explain the results obtained from the STEM and AFM images analyses (figs 7, 11, and 12) more thoroughly.
Author Response
Reviewer 2 suggests that “Authors should describe the novelty of the work and the difference between this work with other similar research.”
We add a sentence in the introduction to defend the novelty of our works: . “However, to the best of our knowledge, the effect of different fumed silica (bearing differ-ent surface energy) on the morphology of PLA/PA11 blends was not studied.”
“The language used in the manuscript can be more specific to the scope and aim of the study.”
We tried to adapt the language (vocabulary) to the present subject that deals with the microstructure and rheology of polymer blends nanocomposites. However, we suggest that Reviewer 2 should be more precise and point out the language elements that he thinks are not dedicated to the scope and aim of the study.
The methodology section is not well organized for the readers to understand the concept.
Reviewer 2 should again be more precise. However, a sentence was added at the end of the Experimental part to explain our strategy: “The methodology of the results Part (below) consists in proving first that the amount of silica of all formulations was effectively 5wt%. This is necessary to compare rheological tests performed on each formulations. Secondly, the oscillatory shear rheology was con-ducted on neat polymer nanocomposites to investigate more about the dispersion (and af-finity) of each silica in each polymer. Then, polymer blend nanocomposites were investi-gated: their rheological behavior, their microstructure. And microstructures were com-pared to the predicted ones (obtained by calculating the wetting parameter). Then, all those results were discussed based on the literature. “
The conclusion section is very general. Must be rewritten and revised.
It is difficult to understand what is expected by reviewer 2 to improve our conclusion. The first part of the Conclusion draw conclusions from what is seen in the Results part and develop in the Discussion part (with reference to the literature) and the second paragraph presents some outlooks. Despite this lake of precision we tried to rewrite some sentences.
What’s authors target from lines 471-511 in the discussion? did they explain the mechanism of this work?
The objective of line 471 to 511 is to explain our rheological and microscopic results based on the literature. We think it is very important to refer to previous articles to understand more and consolidate our results and explanation. Rheological and microstructure results of polymer blends nanocomposites are determined by kinetics ( process parameters) and thermodynamic parameters (interfacial tension, materials parameters) and there are many articles studying each parameter separately. Hence, it is important to collect all those articles and summarize them regarding our present system.
Hence, we tried to briefly explain the mechanisms of these works (for example: “This phenomenon was described by Liu et al. [13], Pawar et al. [8] and Wu et al. [14] and is due to the NPs that drag the domains of wetting polymer causing their co-continuity formation ». Another example : “This is clearly described by Jalali et al. [27], who observed and explained the formation of an elongated PBAT dispersed phase into 70/30 PLA/PBAT blend filled with 3wt% of nanosilica”) to understand more about the final microstructure of our system.
Page 20 “. However, the authors only mentioned that the blend was melt mixed with an internal mixer at 190°C and 30rpm without any more details regarding the processing route (sequence of mixing, especially to avoid particles incorporation in the phase that melts first (i.e. PP)).” These sentences do not make any sense at all. Please revise for improved clarity in order to better understand the readership.
For more clarity, the sentence was modified : “The authors mentioned that the blend was melt mixed with an internal mixer at 190°C and 30rpm without any more details regarding the processing route.”
The authors should review the other investigation on their study way in the introduction part and finally note the novelty of the article. The introduction part needs to develop. Authors can use below references and related works in the field:
High-performance fully bio-based poly (lactic acid)/polyamide11 (PLA/PA11) blends by reactive blending with multi-functionalized epoxy. Polymer Testing, 2019, 78. 105980.
This reference was already cited as reference 5.
Evaluation of clay and fumed silica nanoparticles on adsorption of surfactant polymer during enhanced oil recovery. Journal of the Japan Petroleum Institute, 2017, 60. 85-94.
This interesting article is difficult to cite on our introduction because our article doesn’t deal with adsorption of surfactant polymer and enhanced oil recovery. Hence, despite the interesting conclusions of this reference of 2017 we decided not to cite it in order to keep coherent in our introduction.
The authors should indicate experimental errors throughout the paper.
Thanks for this comment. All error bars are present on our rheological results (except Figure3 because strain sweep tests were conducted only one time). About TGA, Standard deviation were added in Table 4.
Explain the results obtained from the STEM and AFM images analyses (figs 7, 11, and 12) more thoroughly.
To answer this point. Some sentences were added from line 345 to line 355, then line 391 to line 398. Finally, AFM images are compared to SEM and STEM from line 406 to 417

Reviewer 3 Report
The work of Taguet et al. investigates the structuring and rheology of a polyaminde-polylactide blends filled with silica nanoparticles of hydrophilic and hydrophobic surfaces. The work is done accurately, and the experimental results support the conclusions, however the results are not sufficiently interesting in the opinion of this reviewer, that would warrant the publication in nanomaterials. The paper may be suitable for another journal, polymers or materials, e.g.
Author Response
Reviewer 3 suggests to submit the paper to Polymers or Materials that are, according to him more suitable regarding the topic of our article. Despite their fine analysis, we think that our article is more suitable for Nanomaterials as it is dealing with the effect of nanoparticles (especially fumed silica) onto the microstructure (and rheology) of polymer blends. Hence, the “nano-effect” is crucial. Moreover, it is in the scope of the special issue called: Rheological, Thermal and Transport Properties of Polymeric Nanocomposites
Round 2
Reviewer 2 Report
Thank you very much for your detailed answers. The remarks I have posted were generally meant or suggested to be included in the article, not to be explained to the reviewer. Some of them were just addressed by the Authors in form of answers to the reviewer, but in my opinion, they should be at least partly included in the article. I really would like to encourage the Authors to implement much more details to your article for better understanding not only by materials engineers. The potential readers might not achieve a good understanding of the information provided in the article if it is not well explained. My remarks are meant to help you improve the presentation of your valuable research.
Author Response
As suggested by reviewer 2, in this round 2, we tried to examine carefully all the remarks did during the first round. Hence, we tried to do our best to take into account all the remarks and implement much more details for better understanding. You can find the changes done in the first round in red and those performed in the second round in green. There is only one remark very difficult to take into account according to us: “The language used in the manuscript can be more specific to the scope and aim of the study.” We encourage reviewer 2 to give some examples regarding the language that is note specific. Indeed, as our discussion is mostly based on references from the literature, we based our language on those references.
You can find below compiled responses to reviewer 2 from the 1st and 2nd rounds. Please see the attachment.

Reviewer 3 Report
2021.06.11
Authors provided arguments to support their work for the journal Nanomaterials. As the results presented in the manuscript are sound and the paper written correctly, I can recommend it for publication.
There are several minor issues mainly with formatting, that can be improved.
Minor remarks
after sentence in lines 446-447 , the reference number can be mentioned again (ref23 Feng )
line 543 the appropriate references should be mentioned here (again)
lines 346 and 350 formatting to be improved, the figure placed somewhere else and the bold / non bold shrift of the mentioned figures unified.
line 311 “Figure 4” should be moved to the preceeding paragraph.
Figure caption should be given for the figure appearing here.
The figure 4 seems to be present twice in the paper.
Figure 3 is is present twice, too.
line 270-272 displace the figure from the paragraph.
line 220-223 the English style can be improved.
Author Response
Please see the attachement

Round 3
Reviewer 2 Report
accept